# Belonging through meaningful activity in the transition from unhoused to housed

**Patti Plett** [1] *, **Rebecca Gewurtz**[2], **Abe Oudshoorn**[3], **Cheryl Forchuk**[4], **Carrie Anne Marshall**[1]

1 Social Justice in Mental Health Research Lab, School of Occupational Therapy, Western University, London, Ontario, Canada, 2 School of Rehabilitation Science, McMaster University, Hamilton, Ontario, Canada, 3 School of Nursing, Western University, London, Ontario, Canada, 4 Lawson Health Research Institute, Western University, London, Ontario, Canada

☯ These authors contributed equally to this work.
* pplett@uwo.ca

**Data Availability Statement:** I have uploaded the coding tree as a Supporting Information file which includes all the codes used for analysis. In this

## Abstract

### Background

Belonging is closely associated with well-being, yet individuals with experiences of being unhoused are likely to experience social exclusion and challenges with developing a sense of belonging. Engagement in meaningful activity has been linked to belonging; however, there are no focused studies exploring experiences of how engaging in meaningful activities influences belonging. Meaningful activities are things we do that bring value to our lives.

### Purpose

To explore how engaging in meaningful activities may influence experiences of belonging following homelessness through a secondary analysis of qualitative interviews.

### Method

Using interviews conducted in a community-based participatory action study exploring the transition to housing following homelessness (n = 19), we conducted a thematic analysis using the method described by Braun and Clarke. Participants were recruited through communication with local organizations supporting individuals with lived and living experiences of being unhoused as well as through presentations at drop-in organizations. An intentional effort was made to recruit diverse participants regarding housing status, age, and gender. Inductive analysis was used to conduct initial coding, focusing on belonging and engaging in meaningful activities. We then analyzed the codes abductively, using Bourdieu's Social Capital Theory to inform this analysis.

### Findings

The overarching essence generated in our analysis was: "I don't feel like I belong. . .everything in the world is not for me. . .it's for people with. . .enough money to. . .enjoy those things". Within this overall essence, we generated three themes: 1) Human connection: "being where I am with people who care about me, I actually feel good"; 2) Social exclusion:

way, the data is available and without any possibility of identifying the participants.

**Funding:** This study was funded by the Canadian Institutes of Health Research in the form of a Project Grant awarded to CM in 2019 [Grant number: PJT 166132]. The funders had no role in study design, data collection and analysis, decision to publish, or preparation of the manuscript.

**Competing interests:** The authors have declared that no competing interests exist.

being a "regular member of society"; and 3) Non-human connection: "my cats. . .are like my kids to me." Participants described numerous contextual factors that challenged them as they sought belonging following homelessness, including financial limitations and other societal factors.

## Conclusion

Our findings suggest that meaningful activity was an important pathway to belonging for participants in this study.

## Introduction

Individuals with experiences of being unhoused frequently experience social exclusion and difficulties with belonging [1–4]. Additionally, individuals with experiences of homelessness have a higher incidence of mental illness, substance use disorder, physical health issues, and disability [5], creating possible barriers to experiencing belonging. Factors influencing the extent to which a person experiences belonging following homelessness include access to places and activities that promote belonging and the stigma of mental illness, substance use disorder, homelessness, and poverty [6,7]. In a recent systematic review and meta-aggregation of qualitative literature, one of the central themes identified was "belonging through engaging in meaningful activities" [7]. Although several researchers suggest meaningful activities as a possible strategy to foster the development of a sense of belonging and recommend further research to explore this link, few studies have focused specifically on this topic [8–10].

### What is belonging and how does it relate to health and well-being?

Belonging includes feeling valued, respected, accepted, and included [11–14] and is associated with health and well-being [11,15,16]. Patterson et al. [17] noted that "even when someone receives good-quality housing, poverty and lack of meaningful activities can still dictate their daily choices and sense of belonging" (p. 609–610). The environment can impact belonging by directly affecting the availability of choice and the power to choose activities that promote belonging due to perceived, actual, and internalized societal stigma and discrimination [12].

Wong and Solomon [14] proposed a definition of community integration (CI) where contextual factors such as sociodemographic factors, health, functioning, and housing arrangements impact one's ability to integrate into the community. According to this model, there are three dimensions of integration: physical, social, and psychological [14]. Physical integration involves spending time in the community outside of one's housing, social integration involves social interactions and developing a healthy social network, and psychological integration denotes having a sense of belonging in one's community [14]. For this study, we used the definition of psychological integration to define the concept of belonging:

> "the extent to which an individual perceives membership in their community, expresses an emotional connection with neighbours, and believes in their ability to fulfill needs through neighbours, while exercising influence in the community" [14].

> Within this paper, 'community' and 'neighbours' refer to others with whom the individual connects, including those who live within and beyond the immediate neighbourhood where one resides, family members, and persons with whom they connect virtually.

## What is meaningful activity and how does it relate to health and well-being?

Meaningful activities are things we do that bring value to our lives, where the meaning is subjective for each person [18–20]. An applicable definition is: "all the everyday things we do in our life roles, but also the things we do to create a meaningful life and to engage with wider society and culture" [21]. Several aspects denote the concept of 'meaningful,' such as experiencing joy and pleasure, a sense of purpose, and meeting personal needs where individuals can experience choice, control, and belonging [18,22]. Meaningful activities can bring a sense of connection to oneself, to others, and to the environment [19] and can positively impact mental, physical, and psychosocial well-being and life satisfaction [23]. However, when individuals lack opportunities to engage in meaningful activities, they can experience boredom, depressive symptoms, and diminished physical well-being [23,24]. Of particular significance for individuals transitioning out of homelessness is that there is a high risk of experiencing boredom when individuals are unhoused, which may increase poor mental health, substance use, and hopelessness [24].

Individuals exiting homelessness often experience barriers to engaging in meaningful activities such as living in low-income, a lack of access to transportation, and ongoing social exclusion [6,7,25,26]. Living in low-income limits opportunities to engage in activities in the community, including accessing public transportation, unless there are free or low-cost options available [7,27]. Those living with substance use disorder, mental illness, and physical disabilities face additional challenges to engaging in meaningful activities. For example, individuals experiencing mental illness symptoms may struggle to reconnect with friends and family [28] and those in substance use recovery may be rebuilding their social support network while avoiding going to places where they may be triggered to use substances [29]. Going out into the community to engage in activities is difficult for some individuals living with mobility impairments due to getting exhausted quickly, experiencing difficulty getting around the community, or not being able to get out of the house [30].

## Social Capital Theory (SCT)

Bourdieu's SCT guided our interpretation and analysis of findings in this study. Individuals transitioning into housing from homelessness experience ongoing marginalization and exclusion [4,31] and struggle to develop a sense of belonging [4,7]. SCT is useful for understanding the social dynamics that influence power and privilege in society and whether an individual is seen to belong in their community [32]. It emphasizes how societal power dynamics determine the power and privilege a person *may have access to* in society. Additionally, societal perceptions and ideologies (habitus) influence how people act and how they perceive social structures (fields) [32,33].

SCT considers capital as a form of currency, including social, cultural, economic, and symbolic [32]. The more capital (resources) someone has, the greater the likelihood of developing connections and experiencing opportunities to help them progress. Social capital includes the relationships and networks a person has, the benefits of those relationships, and the access and ability to mobilize connections to reach goals [32,34]. Cultural capital helps navigate culture and impacts available experiences and opportunities [32]. Economic capital is essential within the social world and refers to money and assets that can be easily converted to cash [32]. Finally, symbolic capital is the recognition or reputation one receives, legitimizing other forms of capital [32]. A person can only increase their social status if recognized as having higher status [32].

SCT can help with understanding how engaging in meaningful activities influences belonging. People build capital by expanding their social networks, investing in their current relationships, attending school, working or volunteering, and spending time with others [32]. When people have access to money, they can pay their rent to maintain housing, providing a base for building a sense of belonging [7]. Additionally, having money allows individuals to engage in meaningful activities where they might meet new people with whom they could develop a sense of belonging. However, individuals transitioning from homelessness often lack economic capital, have a small or uninfluential social network, or lack skills or education, making it challenging to build capital to improve their situation [32]. Understanding these social dynamics can help address the ongoing marginalization and exclusion experienced by individuals transitioning from homelessness and support their development of a sense of belonging in their community.

## The current study

Our systematic review findings suggest that no existing studies focus on how meaningful activity may facilitate experiences of belonging following homelessness [7]. Bourdieu's SCT [32] provides a lens to help understand the experiences of belonging for the participants in this study. Homelessness prevention involves "providing those who have been unhoused with the necessary resources and supports to stabilize housing, *enhance integration and social inclusion*, and ultimately reduce the risk of their recurrence of homelessness" [35]. As belonging includes integration and social inclusion, it is critically important to understand how engaging in and not being able to engage in meaningful activities influences experiences of belonging to direct future research, policy, and practice. To address this gap in existing literature, we conducted a qualitative thematic analysis guided by the question: "How does engaging in meaningful activities influence experiences of belonging for individuals transitioning from unhoused to housed?" Our findings indicate that engaging in meaningful activities is a key component of developing a sense of belonging following homelessness.

## Methods

We conducted a secondary qualitative data analysis to address the research question. The parent study was a community-based participatory research (CBPR) study exploring the conditions needed for individuals to thrive rather than simply sustain their tenancies following homelessness [36]. This study involved qualitative interviews with individuals with lived and living experiences of homelessness, mental illness and/or substance use disorder. An interpretivist epistemology guided this secondary analysis, where the researchers sought to understand how people interpret, find meaning, and derive purpose from their experiences [37].

## Recruitment

The parent study received ethics approval from both Western and Queen's Universities in Ontario, Canada. Participants were purposively recruited from shelters, permanent and transitional supportive housing programs, and drop-in centres that supported individuals with homelessness experiences in two mid-sized Canadian cities, Kingston and London, ON. Recruitment from these settings included: 1) sending emails to the leaders of the health and social care organizations with details about the study and requesting support with the recruitment process; 2) providing presentations to the staff of relevant organizations, encouraging them to provide contact information regarding this study to potential participants; and 3) attending drop-in sessions at local organizations where individuals could ask questions and

learn more about the study. An intentional effort was made to recruit diverse participants regarding housing status, age, and gender.

Possible biases in recruitment include missing individuals who transitioned to independent housing without the support of health and social care organizations, missing individuals who were experiencing hidden homelessness such as couch surfing prior to becoming housed, and an overrepresentation of individuals who seek out support. Also, the recruitment approach of staff at the various organizations inviting potential participants added a layer of possible bias as the staff were making the decisions regarding who would be invited to participate. To mitigate the possible biases, several approaches to recruitment, as noted above, were incorporated to inform the widest range of participants about the study. Full details of the recruitment process are documented in the original study [36].

**Inclusion and exclusion criteria.** As detailed in the parent study, participants were recruited if they were: 1) over the age of 16; 2) unhoused for at least one month or housed for less than three years after at least one month of homelessness; and 3) acknowledged living with a mental illness and/or substance use disorder. In this secondary analysis, we included only participants who were housed following homelessness.

## Semi-structured qualitative interviews

Interviews with the included participants were conducted in private spaces in collaborating community organizations using a qualitative interview guide. All interviews were conducted during the COVID-19 pandemic with safety measures in place to protect the participants and the interviewers. Interviewers read a letter of information about the study and the request for consent out loud and verbally sought consent prior to conducting interviews. Demographic information gathered included age; gender; sexual orientation; race and ethnicity; income; self-reported mental illness; and housing status. To protect participant confidentiality, their chosen pseudonyms were used with applicable quotes in the findings section of the current study. While unconventional, inviting participants to choose their pseudonyms provides a way to experience ownership and a sense of contribution to the study and offers the opportunity to include part of themselves while maintaining a balance of autonomy and confidentiality [38]. All interview components were read aloud to all participants to avoid threats to trustworthiness related to low literacy. Participants received $40 for participating in the interviews, acknowledging the value of their time committed to this contribution. In participatory action research, it is important to ensure all participants know that they are valued and contributing members of the team. Therefore, participants were compensated for their time at a comparable amount to paid research team members. Interviews were recorded digitally and transcribed verbatim, ranging from 10–82 minutes (Mdn = 43; IQR = 27).

## Analysis

In the secondary analysis presented in this paper, we used thematic analysis to analyze interview transcripts [39,40]. We used inductive analysis to conduct the initial coding and themes. We conducted three rounds of coding, focusing on belonging and engaging in meaningful activities, then grouped codes into like categories [39]. The categories were then arranged into themes and refined through discussions. Finally, we identified an essence and further refined our analysis. We used Dedoose, a cloud-based data management program, to organize the data [41] and finally, to abductively code transcripts informed by SCT [32]. We also created a visual representation of the data to help us to better understand the data.

In consideration that this study is a secondary analysis, the first author took additional steps to prepare for analysis to become familiar with the original study (e.g., reading the

manuscript from the parent study, reviewing the interview questions). Two of the authors were involved in the parent study (AO, CM) and were able to provide further context and insights as applicable to the current study. Additionally, because the data originated from a study focused on thriving following homelessness, the first author reviewed three interviews before conducting the analysis to ensure a good data fit [42].

**Trustworthiness.** To establish credibility in this study, the first author read and re-read the transcripts to understand the participants' experiences. All authors have prolonged experience with the population of interest, achieved by their practice and research experiences related to individuals with histories of homelessness. We were intentionally explicit about the link to the primary data set to avoid potential bias when reading the publications [42]. We also engaged in peer debriefing as a strategy to produce a genuine construction of the participant experiences through a subjective lens and were truthful in reporting the findings, including negative case analysis where the voices of participants whose experiences were not consistent with the overall theme [43]. We used thick description to demonstrate transferability, meaning using detailed descriptions and quotes from the transcripts when writing up the findings [43], allowing others to determine the context and fit when considering applying findings to a different setting. To demonstrate dependability and confirmability, we used a cloud-based platform [41] to assist in the process of analyzing and organizing data [43]. Further, we engaged in ongoing discussions and reflexivity throughout the research process [44].

**Reflexivity.** Collectively, our research team has extensive research and practice experience related to persons who are or who have been unhoused. The first author has clinical experience as an occupational therapist focused on community mental health roles supporting individuals who have experienced homelessness. She also has lived experience with a physical disability, homelessness, occasional housing precarity, and navigating various health and social systems. Our experiences inform the analysis throughout.

## Findings

### Sample characteristics

Nineteen individuals participated in this study comprising of 9 men, 9 women, and 1 non-binary individual. Participants ranged in age from 23–62 years old. The majority of the sample was white. Six participants identified as 2SLGBTQ+. All participants were housed at the time of the interviews and endorsed having a mental illness. A summary of demographic characteristics is provided below (Table 1).

### Qualitative findings

The overall essence within the participant interviews was, "I don't feel like I belong. . .everything in the world is not for me. . .it's for people with. . .enough money to. . .enjoy those things." We generated three themes in our analysis that informed the essence: 1) "Being where I am with people who care about me, I actually feel good"; 2) Being a "regular member of society"; and 3) "My cats. . .are like my kids to me." A visual representation of the theme structure (Fig 1) and a description of the essence and themes are provided below. Additionally, the coding tree is provided as a (S1 File).

### Essence: "I don't feel like I belong. . .everything in the world is not for me. . .it's for people with. . .enough money to. . .enjoy those things" [Bruce]

This overall essence expressed through the themes generated in our analysis was that participants struggled to experience a sense of belonging. Participants' financial situations influenced

**Table 1. Demographic characteristics of participants.**

| Demographic Characteristics of Participants | |
| --- | --- |
| Age Range | 23–62 |
| | (Mdn = 35; IQR = 19) |
| Participant Characteristics (n = 19) | n (%) |
| Gender | |
| Men | 9 (47.4) |
| Women | 9 (47.4) |
| Non-binary | 1 (5.3) |
| Sexual Orientation | |
| Heterosexual | 13 (68.4) |
| 2SLGBTQ+ | 6 (31.6) |
| Bi-sexual (n = 3) | |
| Trans (n = 1) | |
| Queer/Other (n = 1) | |
| Race/Ethnicity | |
| White | 15 (78.9) |
| Indigenous | 2 (10.5) |
| Mohawks of the Bay of Quinte (n = 1) | |
| Cree (n = 1) | |
| Black | 1 (5.3) |
| Other | 1 (5.3) |
| Mental illness[1] | |
| Stress and trauma-related | 13 (68.4) |
| Anxiety | 11 (57.9) |
| Substance use disorder | 9 (47.4) |
| Drugs (n = 9) | |
| Obsessive compulsive thoughts | 3 (15.8) |
| Personality problems | 3 (15.8) |
| Psychosis | 3 (15.8) |
| Income source[1] | |
| Disability-related social support (ODSP)[2] | 14 (73.7) |
| Employment or self-employment | 7 (36.8) |
| General social support (OW)[2] | 2 (10.5) |
| Long Term Disability (LTD) through employment | 1 (5.3) |
| Canada Pension Plan (CPP) | 1 (5.3) |
| Other | 3 (15.8) |
| Housing | |
| City | |
| London, Ontario, Canada | 10 (52.6) |
| Kingston, Ontario, Canada | 9 (47.4) |
| Type of housing | |
| Market rental unit | 10 (52.6) |
| Social housing | 5 (26.3) |
| Permanent supportive housing (Cluster site) | 4 (21.1) |

**Note:** Percentage sums do not all equal 100 due to rounding.

[1]Participant frequencies for race/ethnicity and clinical characteristics exceed the total number of participants due to the classification of participants in more than one category.

[2]ODSP (Ontario Disability Support Program); OW (Ontario Works, a general income support program).

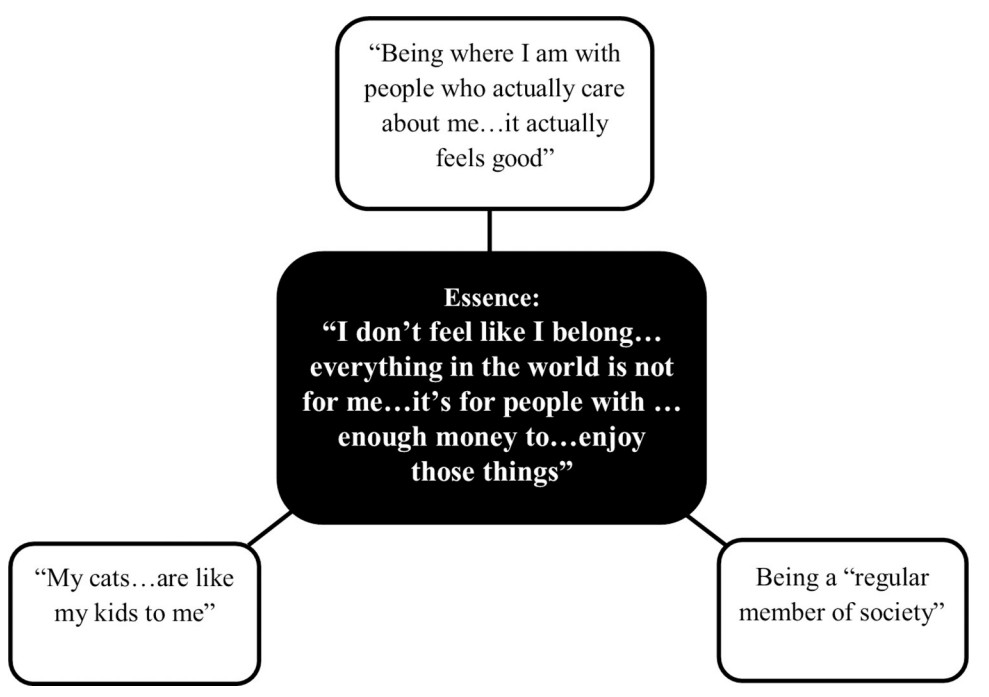

**Fig 1. Visual representation of the findings.**

their ability to engage as a member of society. Most participants were on a low, fixed income and struggled to belong because they could not afford to engage in activities that would enable them to socialize, let alone pay for rent and food. A lack of money prevented participants from having dinner with friends or going to the movies, activities they wanted to do to spend time with others. Pekoe indicated they were feeling "hopeless" because their financial situation "forced (them) to live in yet another toxic living situation," a place where their roommate ate their food, and they could not relax. The "toxic" living environment was a sentiment echoed by numerous participants. Participants suggested increased disability social supports and affordable housing availability as solutions to their financial challenges, enabling them to engage in meaningful activities and address associated difficulties with belonging.

### Theme 1. Human connection: "Being where I am with people who care about me, I actually feel good" [Michelle]

Participants experienced belonging, positive feelings, and a sense of identity when engaging with others who cared about them. They found meaningful connections through investing time in activities with others when creating their own family, renegotiating relationships with their family, and intentionally investing in developing relationships with others in their building. However, many participants had lifelong challenges building healthy relationships.

Family relationships were complex for some participants and required spending time talking or doing other activities together to experience belonging. Some participants formed their own 'chosen family' by focusing on getting to know people they thought might be a good connection for them, developing a strong sense of belonging. For example, Annabella spent significant time intentionally investing in relationships through conversations with others and providing support when needed to create her family.

My family is not someone who I've. . .ever been able to lean on emotionally. . .so I've had to really create my own family. . .It is the closest thing to family. . . that I have. And. . .I can't imagine how to make that better [Annabella].

Nola shared that their siblings needed them to repair the relationships, and in repairing the relationships, they realized their own need to connect with family through engaging in activities with them to experience belonging and well-being:

I think like just to have connections with people or. . .find things that I like to do for me. . .I have to put myself first. Like I used to just do stuff for my kids, like just go to work just come home and be a Mom, it gets exhausting. . .it's not filling my cup. Like I have to go spend time, like me and my sister go shopping, or me and my brother go out for lunch. . .for me those are things that help me [Nola].

Participants identified that engaging in conversations with others fostered a sense of belonging. Casey valued conversing with others in their building: "There's always someone to talk to about our problems. . .the girls are very helpful to each other." Casey added that they do not feel like they belong in the neighbourhood yet, but within their building, "I feel like we're. . .a little commune." Similarly, Annabella spoke about their relationship with one of their roommates:

We might say hi briefly. And that might be it. . .and we might not have full on conversations for a few days or a week. But we could talk whenever we had time. . . We really became a support for each other, and it was mutual, not just. . .one direction [Annabella].

Annabella noted that people leaving homelessness have a need to belong: "Being able to know this is my space, I am welcome and wanted here. Not just tolerated."

Numerous participants found belonging when engaged in activities with others who have experienced similar situations. This was particularly notable for participants with histories of substance use:

I met more people going to concerts or going to outings or going to [local beach] and you're doing it with sober people so like the secondary thing is learning how to do these activities that . . . revolved around alcohol or drugs. [Victor].

Michelle shared that by attending 12-step meetings, they "built a community with the people in recovery." Nola talked about how attending their meetings helped them feel less alone:

I find that. . .my program, my meetings help. . .me get outta my head and. . .to realize like there's a lotta people going through the same thing that I'm going through. Like I'm not alone. . .it's comforting to see. . .a lot of people who are just like you.

By developing a sense of belonging through engaging with others who were also trying to abstain from using substances, they were able to thrive. Similarly, participants felt less alone when they spent time with people who had gone through similar situations as them:

It's crazy how much things changed just because I had that community and I had that support. And it was really multifaceted. Cause part of it was the housing and the mental health and another part of it was meeting people. . .who were also queer and have shared

experiences and then building the confidence to like finally acknowledge my feelings and come out myself. . .and have support in that [Gavin].

Another participant found that, despite going through different traumatic experiences, bonding was possible through conversations because of the shared experience of "going through hell and back" [Alexander].

Participants also experienced belonging by going out into the community and establishing their supports, engaging in hobbies, sharing meals, and hanging out. Getting out into the community was often a low-cost option that helped build a sense of belonging. Barbara shared:

I feel like I belong because. . .in my community where I live, I know my neighbours, . . .I have friends, . . .I go into. . .the stores that I regularly go into, . . .my doctor's office. . . I am setting down those roots here. . .I totally feel like this is where I am supposed to be.

Barbara acknowledged it wasn't easy, "I have to force myself to get out of my apartment and keep those connections with people." Michelle2 noted, "Getting out and going, playing games with my friends keeps me connected."

Sharing meals together often incorporated sharing one's home with others: "There's not a lot of Sunday dinners. . .and that, to me, is connection, doing meals together" [Casey] and "When you have furniture or cutlery and plates and dishes and stuff like that, you can have your friends over, you can have a meal, you can sit and you can chat" [Michelle2]. Gabriella spoke about how hosting Thanksgiving dinner made her apartment "start to feel like home."

Multiple participants faced challenges with being able to engage in meaningful activities and routines, negatively impacting their experiences of belonging at home. Pekoe described this challenge as, "You never really know anybody until you move in with 'em." They had so much anxiety and anger because of how their roommate behaved that they could not sleep or eat and wondered whether they would need to "throw" their roommate out or return to the streets. Participants reported experiencing conflict at home, feeling "watched" [Alexander], and observing fights, theft, and break-ins where they lived [Doc]. These individuals felt unsafe engaging in their regular routines, felt hopeless, and did not experience belonging at home.

Several participants lived in housing far from their friends. Annabella "missed being able to connect with people who are important in [their] life as much because of the distance" and described being far away from their friends as "quite harmful." They typically cycled everywhere, and the commute was too far for them due to their brain injury. Mr. T struggled with loneliness when he got housing because he could not engage in activities with their friends: "Make new friends, make a new friend at (community organization). And now no friends." Gavin discussed the difference living in a location that works for them made on their well-being and ability to connect with others:

I'm able to be social a lot easier because I live downtown. . .I live in the core and a lot of my friends live in the (name of city park) area. Um, so it's really, really nice to be here and not in the West end. Because I feel like, when I've lived. . .closer to the West end, because I don't drive, it's been very isolating and very depressing.

For participants with health challenges, using transit to run errands was very difficult. As a result, they missed opportunities to experience informal interactions with others that could lead to a sense of belonging. A few participants had the luxury of an express bus route near their homes, making commuting and getting around the city easier and increasing their opportunities to belong.

The Covid-19 pandemic also impacted participants' experiences of belonging. One participant described, "Social connection is missing right now" [Doc]. Mr. T talked about the pandemic restrictions preventing them from getting what they needed, "friendship and three meals (a day)" due to the cancellation of communal meals at the community organization. They also spoke about losing soccer, floor hockey, and baseball, significantly changing their routine and limiting opportunities to connect with others. Michelle2 talked about their challenges connecting with others during the pandemic:

With family. . . I'm only able to do it over Skype right now. . . With friends, not so much but it's still. . .a little hard because of everything that is going on in the world right now.

Bruce talked about their disappointment regarding activity-based groups at the community organization being on hold: "I was excited for that to start, but um, it's during Covid that's all sort of not happening." To increase opportunities to engage with others and experience belonging, Michelle suggested that community organizations should have "more events where the community can come together." However, they noted it would be hard to do with the pandemic.

## Theme 2. Social exclusion: Being a "regular member of society" [Barbara]

Participants described wanting to be a "regular member of society" by working, volunteering, and having a regular routine. The concept of a "regular member of society" itself is based on perceptions and societal ideologies. Participants discussed how "regular" roles and corresponding meaningful activities enabled them to experience a sense of belonging within mainstream society. However, disability, health-related issues, finances, location, and transportation access influenced opportunities to participate in society and experience belonging:

Having a brain injury, it's, um, my energy is challenging to manage. And so even if I live close to friends, it's difficult to schedule something in advance. Um, in the midst of a busy school semester because I, it's hard to predict my energy level [Annabella].

"Access" was a recurring theme among the contextual factors that affect activities related to being a "regular" member of society. Participants wanted to engage in activities perceived as part of daily society and felt like outsiders when they could not: "being able to make money is so hard that it just seems like I'm not, it's not a part of the world that I could access" [Bruce].

Employment was identified by participants as a way to feel more like a part of society. Nola stated, "At my work. . .I feel like I'm meant to be right where I am." Another participant described themselves as "I'm a tree planter" and spoke about tree planting and missing this way of life [Alexander]. When asked whether they feel they belong in the community, Mr. T responded, "Yep, I work at the park in the summer." Barbara felt it was helpful that they could continue working and attending their program throughout the pandemic to maintain their connections and support system and, ultimately, their sense of belonging. In particular, Barbara found the continued support she received by attending their substance use recovery program as a stabilizing factor that allowed them to work.

Half of the participants experienced belonging by giving back to the community or helping others. For these participants, giving back was a way for them to take on a meaningful role within society and belong by reciprocating the support they once received [12]. Michelle2 stated, "I belong because I'm able to help my community." Other participants shared this sentiment: "The biggest thing is giving someone hope" [Nola]; and "I'm somebody who likes to be hands-on and, I do a lot of. . .volunteering and giving back to, to the people, you know to

somebody who I once was" [Michelle]. Several participants were either studying or wanting to work in a field such as assisting with navigating the legal system, supporting individuals through their substance use recovery journey, and providing counselling support, where they could give back to the community. They found these supports helpful in their journey and wanted to be able to help others in similar situations. Another way to contribute to society while also building a sense of belonging was by getting involved in advocacy. Annabella said, "I'm very involved with. . . activism. . . and politics. . . and that has really given me a strong community."

Some participants helped in less formal ways and experienced belonging: "It comes down to when someone needs help from me. . .I'm gonna go help them" [Victor]. Nola shared a story where they and a neighbour cooked for each other and developed a beautiful friendship. Bruce reflected on their role as a friend, "I'm a very supportive friend and I'm there for people if they need it." One participant found seeing others struggling hard and wanted to help [Alexander]. Other examples of giving back included being an AA or NA sponsor and providing peer support (formally or informally). However, Casey recalled, "I had friends, and we all hung. . . we helped each other. I don't find there's a lot of that. . . I'm not finding it." Also, one participant used to want to give back but now feels like "the poster child" [Gavin]. Overall, though, participants experienced belonging as they got involved in volunteering, politics, and giving back.

Almost half of the participants described the value of having a routine in being able to connect with others. Annabella highlighted that "scheduling appointments first thing in the morning" made it possible to connect with people. Bruce recalled, "it's nice living with friends. . .when we share responsibilities." Barbara shared how a routine helped them feel part of the community through jobs, school, and engaging with others. When the pandemic hit, they lost their routine and connection with the community:

> Routine is big for me. When Covid hit, I went through a great depression. . .I am very institutionalized and need things to keep going the way that they are [Barbara].

Another way routines can be helpful is to help make one's place feel like home: "I like to sit upstairs by my window. . .and just read my book, do my journal, drink a coffee, have a beer" [Alexander]. Alternatively, some participants did not experience belonging. Amber described, "just sitting there not doing nothing. . .I'm wanting to do something. I mean, I wanna do drugs." Amber added that they want to go out into the community and find a hobby, acknowledging that they feel alone. Overall, having a regular routine provided the opportunity for participants to be able to spend time with other people and engage in meaningful activities.

Participants experienced long-standing barriers to belonging and community engagement, often due to disability, mental illness, and traumatic histories. One participant described this challenge: "Most of the time I just feel like I'm on the peripheral. . . I'm not even certain sometimes I'm on the edge" [Gabriella]. Annabella shared their struggles with feeling like they could not be authentic with others because they had to keep their history of being unhoused "a secret." Ocean Breeze talked about wanting to live "in the middle of nowhere, where nobody can hurt me, so I can't hurt anybody else." They continued, "everything tells me I need to be here because I need support, I need help."

Participants talked about depression, anxiety, and paranoia affecting their ability to work or even leave the house. Casey disclosed that having conversations was hard for them: "My mind races a lot from my bipolar, so trying to keep up, like, a conversation in my mind at the same time is hard." Casey also reflected, "I don't trust. . .Not that people are bad. Just. . . I guess I hurt too much losing all the longer-term ones. . .It's just hard making friends." Amber talked

about not knowing what it would feel like to belong and that they did not belong because of "the way I act sometimes, I guess." Another participant shared that they struggle with daily life:

> I have PTSD. . .I've experienced a lot of trauma over a lot of years and a lot of continued trauma. . .which has affected me in my adult life where it's hard to find work and it's hard to function as an adult and as a person because of that [Gavin].

Several participants identified that isolation and living alone were detrimental to their mental well-being. For some individuals with a substance use disorder, their substance use got in the way of wanting to get out of the house, making it challenging to develop and maintain healthy relationships. Michelle2 reported, "if you sit around in your apartment all day, you're just gonna get depressed." They added, "if you keep occupied. . . it gives you the motivation to do other things because you're out and about." These strategies also help with building a sense of belonging.

Individuals living with physical disabilities, physical health issues or cognitive disabilities faced physical barriers to belonging. Navigating the city using a mobility device was challenging for some participants, as uneven surfaces could cause their device to get stuck in a sidewalk crack. Gabriella stated that people with mobility challenges "won't go out. . .because it's just too hard. . .and winter is coming," resulting in less social contact with others. They perceived that society sees people who are seniors or who have a mental illness, physical disability, or a fixed income as a "drain on the system. We're just taking up space." The physical challenges of leaving the house, such as climbing stairs, made everyday tasks difficult for others. Pekoe reported avoiding basic everyday tasks due to difficulty walking and constant pain, including getting food and other essentials. These challenges were compounded by the "psychological ramifications" of their life situation, making it difficult for them to engage with others and experience a sense of belonging:

> If you think about the psychological ramifications of your, your history and everything else that's going on in your life right now. The homelessness. . .that's such a bundle to handle all the time. . . And then having to deal with that on the same side with the pain I deal with every day. . . I fly off the handle a lot but you know it's just. . .I don't know what to do sometimes. . .I'm just. . .at my wits end. . .I just wanna give up and say fuck everybody and just jump on a bus and just leave. . .I just feel unwanted, uncared for, and nobody really gives a shit.

### Theme 3. Non-human connection: "My cats. . .are like my kids to me" [Gavin]

Participants described a sense of belonging through interacting with the living world around them, including pets and nature. Several participants had a pet and perceived it as a family member, creating a strong bond and sense of belonging. Casey stated, "I consider them my family. . .they're really important." Gabriella spoke fondly of their pet:

> That little furball. . .I'm as much her mom as if I'd given birth to her. . .she knows she can count on me. . .When you have the responsibility of another life, you have to take care of it.

Gabriella, Casey, and Gavin shared their experiences of caring for their cats, giving them a sense of purpose and belonging. Gabriella spoke about the reciprocal element of belonging with their cat, where their cat trusted and depended on them, and by providing for the cat's

needs, there was a mutual relationship. Similarly, Casey shared that "after losing everything like I did, having the responsibility of something to make sure they're fed every day and watered, and making sure I budget properly to feed them" gave them a sense of purpose and belonging. She likened their cats to humans, describing one of them "like a little old man." Casey found comfort in her cats, "when I am agitated, I can pet them and it calms me. . .they're good company." Victor shared how having a dog was good for their mental well-being and that they, their partner, and their dog make a "nice family." Gavin's new apartment "was a place where (they) could take care of (their) cats." In particular, Gavin had previous places where they were not allowed to have a cat or where the landlord caused problems because they had a cat. For each participant, taking care of their pets was a meaningful activity that created a strong bond and a sense of belonging with their pets.

Participants also experienced a connection and a sense of belonging when spending time in nature. Ocean Breeze experienced a sense of belonging when connecting with nature but struggled with motivation to do anything due to their substance use:

> I go to the water a lot and sit down by the water, that's about it. I don't know, I have no motivation to do anything anymore. . . it's like crystal meth took it all away. . . I was full of life. . .I used to go on nature walks, I'd be in nature a lot.

When living in an encampment, Ocean Breeze found a sense of connection with nature and feeling "grounded." Casey noted, "I sit outside, like that's my way of getting in tune with nature," and spent significant time beautifying her backyard. Spending time outside was a big part of Alexander's life, who enjoyed several outdoor hobbies, including longboarding, cycling, and playing his uke, and identified themself as a tree planter–an identity strongly linked to nature. One of the ways Sunshine connected with the community was by going for a bike ride. Gavin spoke about how living downtown, where they had access to being in nature, helped them feel at home:

> I'm really avid on cycling. . .When something bad was happening at home I would get outside and cycle and that would make me feel better. . .but also being able to, you know, bike downtown or you know, walk to work or um, just go for a ride um, and being near parks and being near shops and not have to bus everywhere.

Annabella emphasized the importance of being outdoors, connecting with nature, and rejuvenating their mental well-being:

> And even just walking outdoors. Or even just having a nice backyard. . . there's simple things like that where. . . that is an urban planning thing for sure, but it's like. . .if we're designing a space it has got to have like safe spaces that. . . that provide green space and trees and like we, we need nature. Like that is a need not just a like, oh this is nice and this is frivolous. Like we need that.

Participants highlighted that finances and opportunities impact having pets and interacting with nature. Gavin appreciated having a place where they can care for their cats, which spoke to the challenging aspect of obtaining housing that allows pets. Participants expressed the difference between the provincial fixed income supports, the general support income and the disability support income. On the disability support income, they had enough income to have pets, but on the general support income, they would have to give up their pets to live somewhere affordable because of associated costs. Gabriella disclosed that they prioritize their

"kitty" over their own needs when money runs out. Casey sacrificed treatment (for substance use disorder) because they could not stay longer than 28 days as their cats would not do well with a long separation. Annabella indicated that money is often needed to be able to access green spaces:

> Rich people buy places or rent places close to those things [green spaces, trees, nature] . . .Whereas. . .if you're experiencing poverty and you don't have a choice you have to move into a place where like it's all concrete or it's, it's an apartment building and there's not good parks nearby. Or they're too far to walk to or it doesn't feel safe to walk there.

## Discussion

Belonging is a critical outcome for individuals when they transition to housing as when individuals feel included and become integrated they are more likely to remain stably housed [35]. Researchers have proposed a link between participating in meaningful activities and belonging following homelessness [8,10,17]; however, in a systematic review focused on belonging following homelessness, few studies focused on meaningful activities and no studies focused on experiences of belonging [7]. We designed the current study to address this gap by exploring how engaging in meaningful activities may facilitate belonging for individuals transitioning from unhoused to housed.

Our findings indicate that participants struggled to develop a sense of belonging. Their financial situations impacted their ability to participate in meaningful activities that would facilitate belonging in their communities. Participants experienced belonging when engaging in activities with others who cared about them, when they were participating in activities as "a member of society," and by engaging in activities that enabled them to interact with the living world around them. However, multiple contextual factors limited opportunities to belong, such as living in low-income, challenges with reconnecting with family members and friends, and health and disability-related factors that made it difficult for participants to engage in activities within the community where they would have more opportunities to meet people.

### Applying the SCT lens

Through the lens of SCT [32], the experiences described by participants in the current study indicate that they lacked the social power needed to engage in meaningful activities to develop a sense of belonging. Engaging in activities with others would allow them to build social capital and, in so doing, increase possibilities for gaining access to further opportunities. However, participants experienced social exclusion and stigma due to their history of homelessness, mental illness, and poverty, resulting in limited social, cultural, and economic capital. These findings are consistent with the literature identifying that individuals continue to experience social exclusion and stigma based on their histories of homelessness, mental illness, and poverty [4,17,45,46]. Some participants in our study struggled with boredom and substance use and wanted to find hobbies in the community to keep occupied. The reduced access to support during the pandemic hindered their access to social capital. Similarly, in a study focused on formerly homeless individuals living in Permanent Supportive Housing, participants struggled with loneliness and isolation due to the cancellation of activities and programs and limited access to shared spaces because of the pandemic, negatively impacting their ability to connect with other people [47].

Participants in our study desired to be "regular members of society" and saw this as one aspect of belonging. The value participants placed on employment, giving back, and having a

regular routine as determinants of being part of society demonstrates the influence of one's habitus, or the societal values and ideology that become internalized [33,48]. Despite the internalized ideology, participants found meaning through working or giving back, building their social capital and inner confidence (cultural capital) and facilitating further opportunities for connection [32]. They also found a sense of identity and were furthering their skills, adding to their cultural capital. Likewise, formerly homeless participants in one study experienced improved self-esteem, a sense of identity, belonging, and "what they did mattered" [49]. With the increased capital in a few areas, the participants were more likely to experience additional opportunities [32].

Participants' health and disabilities negatively impacted their ability to build social capital. Some perceived themselves as a "drain on society," impacting their self-efficacy and capacity to build connections with others. If participants had better access to social and economic capital, they could decide where they live, afford suitable housing in neighbourhoods where they can engage in meaningful activities, and have a greater likelihood of experiencing belonging.

Strategies for engaging with the living world include gardening and various arts-based activities such as using materials from nature to create are or painting scenery. Community gardening has the potential to meet several needs for individuals as they transition from homelessness. Gardens provide a source of low-cost and healthy food as well as opportunities to build connection and community and a sense of ownership [50]. Hamilton et al. [50] developed a community-based program for individuals who transitioned out of homelessness that included groups where individuals built social connections, received resources such as a community-made cookbook, had access to kitchenware, and grew their own garden produce. Engaging in creative arts has been shown to be beneficial for individuals with living experience of homelessness, providing opportunities to connect with their culture, as a way to connect with oneself, and as a way to share experiences and explore spirituality [51]. During homelessness, the lack of housing was a barrier [51]; however, it is possible that individuals who are recently housed may experience similar benefits and have a space they can engage with these activities on their own as well as with others as way to experience belonging.

## Research implications

Future research opportunities identified in this study include understanding how engagement in meaningful activities facilitates belonging following homelessness. Building on previous research exploring experiences of belonging following homelessness [7], many participants struggled to develop such a sense of belonging in this study. Having activities available within one's building or community may facilitate the development of a sense of belonging [9]. However, similar to the findings by Marshall et al. [27], limited income impacted opportunities to engage in meaningful activities for participants in this study.

The location of participants' neighbourhoods, the ability to engage in meaningful activity, and belonging were linked in this study. Participants experienced loneliness and isolation when they lived far from their friends and support system because they could not engage in meaningful activities with others easily. Therefore, they desired to live near the city core for better access to meaningful activities, support, and friends. However, some participants felt trapped because they could not afford to live elsewhere. Ecker and Aubry [52] found that the neighbourhood facilitates CI when it has good amenities nearby and provides good social opportunities. Further research is needed to explore the relationship between the neighbourhood's location and its amenities (e.g., access to public transit, places to connect with others), engaging in meaningful activities, and belonging within the new community for individuals transitioning out of homelessness.

Additionally, access to opportunities to engage was a factor in whether participants in our study experienced belonging. La Motte et al. [45] found that belonging was the most significant factor in the process of CI and for mental wellness. Yet, due to interpersonal factors, one's perception of their place in society, and the availability (or lack thereof) of opportunities to engage with other neighbours in the community, it is one of the most challenging elements to address in the transition from unhoused to housed [45]. Further research is required to better understand how social roles influence opportunities to engage with neighbours during the transition from unhoused to housed.

## Practice implications

Health and social care providers supporting individuals transitioning from unhoused to housed should consider how current practices and programs impact the sense of belonging for these individuals through engagement in meaningful activities. Access to free activities and programs at community organizations was a contributing factor in experiencing belonging for participants in this study. Service providers are encouraged to intentionally develop programming that has the potential to facilitate belonging when housed, such as peer support [53], therapy groups [9], and activity-based groups like yoga or art [9]. Providing community events like festivals can also support belonging [54]. Access to activities should be free and low-barrier, given that poverty is an ongoing issue for individuals following homelessness [25]. Service providers should advocate for free or low-barrier access to recreational activities within their city to support belonging for individuals transitioning from homelessness.

## Policy implications

The current study identifies an overarching theme of participants struggling to belong due to their fixed income, limiting where they could live, opportunities to access meaningful activities, and their capability to belong in society. Research supports this finding, noting a connection between social exclusion and low income for individuals with mental illness [55]. The lack of free options for activities and affordable transportation prevented participants in our study from socializing and building social capital, negatively impacting their development of a sense of belonging. Patterson et al. [17] note that even with suitable housing, low income and reduced opportunities to engage in meaningful activities can impose restrictions on daily choices and impact belonging. Similarly, Marshall et al. [8] describe the process of struggling to meet basic needs above the cost of rent as "negotiating survival" (p. 40). Kerman and Sylvestre [56] report that limited incomes negatively impact the ability to engage in recreational activities and that programs aimed at reducing the cost of bus fares allow individuals to travel within the city. There is a need to advocate for offering free or low-cost activities and supports and, where possible, implement these strategies. Additionally, there is an urgent need to address income inequality through initiatives such as a Universal Basic Income (UBI) or increase the payments within existing income support programs. Housing insecurity can be addressed by developing permanent and significant rental subsidies, and eviction prevention programs [57,58]. These strategies would increase opportunities for individuals to engage in meaningful activities, pay for them, and positively impact their potential for developing a sense of belonging.

## Limitations

In our study, the majority of the participants were White, with only two individuals identifying as Indigenous, one as Black, and one as 'other.' Indigenous persons represent a small portion of the Canadian population [59] but make up a significant portion of the urban unhoused

population [60], while racialized groups represent 28% of individuals experiencing homelessness [61]. Therefore, the findings of this study may not be transferable to racialized groups due to their unique experiences. However, it is important to consider why representation from non-White groups, and specifically from Indigenous persons because Indigenous persons make up such a high proportion of unhoused individuals in Canada. Typical approaches such as Housing First have not supported Indigenous peoples well [62]. It is possible that, for similar reasons, potential Indigenous participants may not have been involved with the community organizations that were approached for recruitment purposes. However, Indigenous researchers [60] recommend collaborations between organizations supporting Indigenous persons and academic researchers to ensure the needs of Indigenous peoples are met and that there is funding available to conduct the research. Additionally, studies focused on Indigenous peoples should include Indigenous persons as part of the research team [63].

The interviews were conducted during the COVID-19 pandemic, which may have affected participants' time used and their ability to engage in meaningful activities, potentially impacting their experiences of belonging. Also, the pandemic may have prevented some individuals from contributing to the study, thus impeding valuable contributions from such individuals. Additionally, access to technology was not a focus, but some participants mentioned challenges accessing virtual medical appointments or communication platforms. Virtual environments can be a place of belonging and support, so this is an important consideration.

The study was conducted in Canada, so researchers, service providers, and policymakers outside of North America should be aware of the context. Secondary analysis has potential limitations, such as data fit [42] and authors not being present during data collection [42,64], but the authors addressed these challenges by screening interviews for data fit and immersing themselves in the data through reading and re-reading [39].

## Conclusions

Our findings suggest that poverty influences one's ability to participate in meaningful activity following homelessness and, in turn, their opportunities for belonging in their communities. Being limited in this way prevents one from building social capital that can support thriving following homelessness. Money is required to pay rent, buy food, use public transportation, and socialize. On a limited income, participants in our study struggled to make ends meet, let alone afford to participate in activities with others. Engaging in meaningful activities was a facilitator in developing a sense of belonging. There were three ways in which participants experienced belonging: through spending time with others who care, engaging in activities that help individuals feel like they are part of society, and interacting with the surrounding living world, such as pets and nature. In addition to limited income, numerous contextual factors impeded the development of a sense of belonging, including a history of complex relationships, unhealthy living situations, the location of housing, and disability and health-related issues. Researchers, practitioners, and policymakers should recognize the critical importance of belonging in supporting thriving and preventing ongoing homelessness in future research, and the ways in which contextual factors mitigate this experience.

## Supporting information

**S1 File. Supplemental information: Coding tree exported from Dedoose.** This file (S1 File) provides the coding tree for the data, including the parent ID, depth of coding level, and code name (title). The data was exported from Dedoose [41].
(XLSX)

## Author Contributions

**Conceptualization:** Patti Plett.

**Formal analysis:** Patti Plett, Rebecca Gewurtz, Carrie Anne Marshall.

**Methodology:** Patti Plett.

**Supervision:** Carrie Anne Marshall.

**Visualization:** Patti Plett.

**Writing – original draft:** Patti Plett.

**Writing – review & editing:** Patti Plett, Rebecca Gewurtz, Abe Oudshoorn, Cheryl Forchuk, Carrie Anne Marshall.

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
