## [Decision Letter · Decision Letter 0]

23 Jun 2024

PONE-D-24-09042Belonging through meaningful activity in the transition from unhoused to housedPLOS ONE

Dear Dr. Plett,

Thank you for submitting your manuscript to PLOS ONE. After careful consideration, we feel that it has merit but does not fully meet PLOS ONE’s publication criteria as it currently stands. Therefore, we invite you to submit a revised version of the manuscript that addresses the points raised during the review process.

If you could please address the reviewers comments as per below.  

We look forward to receiving your revised manuscript.

Kind regards,

Katie Gibson Smith

Academic Editor

PLOS ONE

Reviewers' comments:

Reviewer's Responses to Questions

**Comments to the Author**

1. Is the manuscript technically sound, and do the data support the conclusions?

Reviewer #1: Yes

Reviewer #2: Yes

2. Has the statistical analysis been performed appropriately and rigorously? 

Reviewer #1: N/A

Reviewer #2: Yes

3. Have the authors made all data underlying the findings in their manuscript fully available?

Reviewer #1: Yes

Reviewer #2: Yes

4. Is the manuscript presented in an intelligible fashion and written in standard English?

Reviewer #1: Yes

Reviewer #2: Yes

5. Review Comments to the Author

Reviewer #1: Manuscript generally written well. My comments are mainly minor.

Abstract: meaningful activity not defined.

Aim: please use SPIDER framework to write aim for qualitative studies. Also in the main text.

Method: provide further details about the design, participant identification and recruitment, sampling etc.

Results: perhaps provide theme headings as it is difficult for readers to make sense of key results from the ‘essence’ presented.

Main text: Also define meaningful activities here. In the background mention resource related barriers to engaging in meaningful activities plus the importance of comorbidities.

In the methods, please describe study settings, participant recruitment and consent processes. Referring to the parent study is not enough for the readers.

Explain the high reimbursement rates.

What if participant chosen psuedonyms would identify them? I have not seen this practice before.

Suggest avoid referring participants with ‘his’ ‘her’, use neutral tone (e.g. their) to further ensure anonymity.

On the theme ‘being a regular member of the society’- please describe if there was any roles or scope for healthcare professionals that participants identified which would facilitate this process. Perhaps this reference can help https://doi.org/10.3399/bjgp18X694577 to discuss this aspect.

In the discussion, please incorporate some literature on how arts based interventions e.g. arts, gardening could be meaningful activities to engage people who have experienced homelessness- cite some references and learnings available. As this aspect seem to be missing.

Reviewer #2: Thank you for the opportunity to review this manuscript and research. Overall, this has been written to a high standard and is clear and well written throughout. The manuscript engages with an important topic and explores the concept of ‘belonging’ in regards to individuals with experiences of being unhoused. Important and insightful work from those with lived and living experience is presented and the authors have generated rich insight into how ‘meaningful activities’ develop a sense of belonging.

The context and rationale are well-described and the aims of the research are clear with a solid methodology. I have some minor points in regards to the presentation of the themes within the findings and areas for improvement (see below points). The discussion and conclusion provide a robust summary of the research.

Introduction

1. While you have explored the context of ‘belonging’ within the introduction, linking this to health and well-being, the ability to integrate into a community and Bourdieu’s Social capital Theory, it would be helpful to provide more clarity in regards to ‘meaningful activities’ as this is a key aspect of your research objective. Meaningful activities are briefly explored in (lines 101-107), but more detail would enable the reader to gain a sense of your interpretation and definition of what this aspect means within the context of your study.

Methods

2. Recruitment (line 138). It would be valuable to acknowledge any potential sampling bias due to purposive recruitment and what steps were taken to mitigate against this. Addressing this will enhance the credibility and rigour of the study design.

3. Analysis (line 162). The study described using thematic analysis to analyse interview transcripts, drawing from Braun and Clarke (2006). Given that TA is utilised extensively across qualitative research, it would be helpful to provide more detail to explain how this approach was applied within the analysis of this specific study. There are now many different versions of TA available to qualitative researchers since the 2006 publication and acknowledging that would provide further detail to justify your particular TA approach. It would also be useful to provide further clarity in regards to the process used, for example, was this an inductive or deductive approach?

Qualitative Findings

4. Presentation of themes throughout the manuscript (Abstract and Findings). While I found the quotes powerful to convey the themes generated, I would have liked an inclusion of a name or definition alongside the thematic quotes. This would provide greater clarity in formulating your interpretation of each theme to convey explicit understanding of the data. For example, is Theme 1 speaking to the importance of ‘connection’ or ‘relationships’? Is Theme 2 speaking to ‘social exclusion’ or ‘discrimination’? Is Theme 3 speaking to the importance of non-human interactions? The quote for theme 3 highlights the importance of pets, but also discusses nature as an important aspect, but this is not captured in the overall essence of the theme. Without some contextualisation of the themes by defining and naming them, alongside the quotes, the reader is having to interpret this themselves.

Discussion

5. Line 524. This sentence seems to end abruptly when highlighting that multiple contextual factors limited opportunities to belong. It would be helpful to provide more detail in regards to this, what are the contextual factors, a brief summary here for clarity for the reader.

Practice Implications

6. Line 584. More detail in regards to what is meant by ‘service providers’ as this is unclear. Do you mean health and social care services? Community organisations? This will enable you to connect to the context and be helpful for the international audience.

Limitations

7. Line 616-621. It was good to see you acknowledge the limitations within this study, such as the majority of participants identifying as White. However, it would be helpful to provide further rationale in regards to why this was the case in your study and what steps, if any were taken to ensure a diverse sample. What can be done to address this gap within future research?

6. PLOS authors have the option to publish the peer review history of their article (what does this mean?). If published, this will include your full peer review and any attached files.

Reviewer #1: No

Reviewer #2: **Yes: **Dr Natalia Farmer

---

## [Author Response · Author response to Decision Letter 0]

31 Aug 2024

I have uploaded a Response to the Reviewers document, addressing the comments and recommendations provided by the reviewers.

---

## [Editor Report · Decision Letter 1]

6 Sep 2024

Belonging through meaningful activity in the transition from unhoused to housed

PONE-D-24-09042R1

Dear Ms. Plett,

We’re pleased to inform you that your manuscript has been judged scientifically suitable for publication and will be formally accepted for publication once it meets all outstanding technical requirements.

Kind regards,

Katie Gibson Smith

Academic Editor

PLOS ONE
---

## [Editor Report · Acceptance letter]

17 Sep 2024

PONE-D-24-09042R1 

PLOS ONE

Dear Dr. Plett, 

I'm pleased to inform you that your manuscript has been deemed suitable for publication in PLOS ONE. Congratulations! Your manuscript is now being handed over to our production team.

Kind regards, 

on behalf of

Dr. Katie Gibson Smith 

Academic Editor

PLOS ONE